# Longitudinal intravital imaging of the femoral bone marrow reveals plasticity within marrow vasculature

David Reismann[1], Jonathan Stefanowski[1,2], Robert Günther[1], Asylkhan Rakhymzhan[1], Romano Matthys [3], Reto Nützi[3], Sandra Zehentmeier[1,2,6], Katharina Schmidt-Bleek [4], Georg Petkau[1], Hyun-Dong Chang[1], Sandra Naundorf[1], York Winter[5], Fritz Melchers[1], Georg Duda[4], Anja E. Hauser[1,2] & Raluca A. Niesner[1]

The bone marrow is a central organ of the immune system, which hosts complex interactions of bone and immune compartments critical for hematopoiesis, immunological memory, and bone regeneration. Although these processes take place over months, most existing imaging techniques allow us to follow snapshots of only a few hours, at subcellular resolution. Here, we develop a microendoscopic multi-photon imaging approach called LIMB (longitudinal intravital imaging of the bone marrow) to analyze cellular dynamics within the deep marrow. The approach consists of a biocompatible plate surgically fixated to the mouse femur containing a gradient refractive index lens. This microendoscope allows highly resolved imaging, repeatedly at the same regions within marrow tissue, over months. LIMB reveals extensive vascular plasticity during bone healing and steady-state homeostasis. To our knowledge, this vascular plasticity is unique among mammalian tissues, and we expect this insight will decisively change our understanding of essential phenomena occurring within the bone marrow.

---

[1] Deutsches Rheuma-Forschungszentrum, A Leibniz Institute, Charitéplatz 1, 10117 Berlin, Germany. [2] Immune Dynamics, Charité—Universitätsmedizin, Charitéplatz 1, 10117 Berlin, Germany. [3] RISystem AG, Talstraße 2A, 7270 Davos Platz, Switzerland. [4] Julius Wolff Institute, Charité—Universitätsmedizin, Augustenburger Platz 1, 13353 Berlin, Germany. [5] Humboldt-Universität zu Berlin, Unter den Linden 6, 10099 Berlin, Germany. [6] Present address: Department of Immunobiology, Yale University School of Medicine, New Haven, CT 06519, USA. David Reismann and Jonathan Stefanowski contributed equally to this work. Anja E. Hauser and Raluca A. Niesner jointly supervised this work. Correspondence and requests for materials should be addressed to A.E.H. (email: hauser@drfz.de) or to R.A.N. (email: niesner@drfz.de)

The bone marrow is the birthplace of hematopoietic cells in adult mammals. As such, it is a highly dynamic environment, where new blood cells are constantly generated from proliferating hematopoietic precursors and exit the bone marrow into the circulation[1]. At the same time, the bone marrow serves as a harbor for memory cells of the immune system, which reside in the various subtly different microenvironments that support specific immune cell types[2]. These microenvironments are characterized by specialized stromal cell populations, which compose stable components of the niches[3] and provide essential signals for the differentiation and survival of the hematopoietic cells that occupy these niches.

In order to fulfill these functions, the bone marrow tissue is traversed by a dense system of blood vessels comprising arteries, distal arterioles, and sinusoids. These are responsible for transporting cells entering and exiting the bone marrow[4, 5], and also for delivering oxygen, nutrients, and growth factors[1]. The marrow vasculature plays a key role in the regulation of hematopoiesis[6], and hematopoietic stem cell niches are located perivascularly. Recently, a strong link between angiogenesis and osteogenesis, mediated by a defined vessel subtype, characterized by CD31[hi]Emcn[hi] (type H) endothelium has been described in the bone marrow. This finding revealed a previously unknown heterogeneity among blood vessels in the bone marrow, supporting the notion of tight functional interactions between marrow and bone[7].

In the recent decade, intravital two-photon microscopy has significantly advanced our understanding of dynamic processes within the immune system. Within the bone marrow, intravital microscopy has helped to elucidate mechanisms of hematopoiesis[8], mobilization of hematopoietic cells[5, 9], and the maintenance of immunological memory[3, 10]. In mice, intravital imaging of bone marrow in areas close to the bone cortex has been performed either in the calvarium[11–13], in the tibia[3, 14], or in the femur[15, 16]. The calvarial preparation takes advantage of the thin sheet of flat bone covering the marrow in this area. Imaging the bone marrow of long bones is more invasive, as it requires the surgical ablation of cortical bone. Both methods have been used mainly as terminal procedures, although imaging at multiple time points has been used for intravital microscopy of both calvarium[17, 18] and long bones (femur and tibia) for imaging durations of hours, over a maximum of 40 days[15, 16]. Nevertheless, up to now there is no available method enabling longitudinal intravital microscopy of the deeper marrow regions in long bones, at sub-cellular resolution, over the time course of several weeks or even months, i.e., both during bone healing and during homeostasis. In order to understand the cellular dynamics occurring in those marrow regions over longer periods of time, an intravital imaging approach allowing longitudinal observation of a fixed region within the bone marrow in one and the same subject is needed.

The development of permanent windows for the brain cortex[19] or of the spinal cord[20] solved the challenge of longitudinal imaging, but limitations regarding the accessibility of deep tissue areas still remained. An elegant solution for this problem was provided by the lab of Marc Schnitzer[21], who used gradient refractive index (GRIN) endoscopic lenses implanted into the brain cortex in order to image deep cortical layers over several weeks. An endoscopic approach was also used previously for single-photon imaging in the femur, by introducing a fiber-optic probe into the femoral cavity through the knee area. This method

was used for imaging within a single-plane circular field of view of 300 μm diameter and at a lateral resolution of 3.3 μm. The imaged tissue areas were located at 10–15 μm distance from the endoscope tip[22].

Here, we present a novel method called longitudinal intravital imaging of the bone marrow (LIMB), which allows repeated imaging of the same tissue region in the bone marrow of living mice over the time course of up to 115 days. The approach enables sub-cellular resolution multi-photon imaging of cylindrical tissue volumes ($300 \times 300 \times 200$ μm$^3$) and is based on the use of a GRIN endoscopic lens mounted on a specialized holder that is surgically fixated to the femur of the mouse. By 28 days post-surgery, reactive processes of the organism to the implant completely cease, and the tissue reaches equilibrium. Using our technique, we are able to demonstrate a high degree of structural plasticity of deep bone marrow vessels not only during bone healing following lens implantation, but also in steady-state homeostasis, with implications for the concepts of micro-environmental stability and niche formation.

## Results

**Characterization and optical performance of the LIMB implant.** In order to understand tissue and cellular dynamics in the bone marrow on a longer time scale, we developed a biocompatible implant—the LIMB implant, which allows microendoscopic imaging of the femoral bone marrow.

The implant is based on a fixation plate originally developed to stabilize the mouse femur after osteotomy[23]. Based on the principles of low contact plates developed for fracture healing[23–27], stability is achieved via two angle-stable bi-cortical screws (Fig. 1a, Supplementary Fig. 1d), which fix the plate in a "bridging" position above the bone, to avoid direct contact to the bone surface. This prevents any compression of the bone.

In order to allow repeated intravital imaging deep within the femoral bone marrow, a titanium alloy tube of 600 μm outer diameter and 450 μm inner diameter is mounted onto the fixation plate (Fig. 1a; Supplementary Fig. 1). To account for tissue heterogeneity and to visualize the different bone marrow areas of the femur, two types of implants were designed, allowing us to access either diaphyseal or metaphyseal regions (Fig. 1b). By varying the length of the endoscopic tubing in the marrow cavity (e.g., 500 vs. 700 μm, Fig. 1c), we can image pericortical tissue areas or areas deep in the marrow. For instance, by using a long microendoscopic tubing, we can visualize the endosteum on the opposite side of the bone cortex, in a contact-free manner.

A GRIN lens (Fig. 1d) is positioned in the endoscope tubing and used as a lens for imaging. We use two GRIN lens designs,

**Fig. 1** LIMB allows murine long bone imaging in various locations with high resolution. **a** Design and positioning of LIMB implant for longitudinal bone marrow imaging. The LIMB implant is fixed onto the femur using bi-cortical angle-stable screws. GRIN lens systems are placed within the endoscope tubing for imaging and sealed to ensure sterility. The positioner allows adjustment and alignment of GRIN and microscope optical axes. **b** In order to account for tissue heterogeneity, i.e., metaphyseal vs. diaphyseal regions, alternative LIMB designs have been developed. LIMB fixation with four screws allows higher bone stability after osteotomies. **c** Tubing lengths of 500 and 700 μm, respectively, allow access to either peri-cortical or deep marrow regions. **d** Two GRIN lens systems are used for imaging. The single GRIN lens (upper panel) combines the imaging and objective lens function and is glued into the endoscope tubing. The symmetric triple GRIN lens (lower panel) is exchangeable and a sapphire window seals the endoscope tubing. **e** 3D fluorescence image of the bone marrow of a *CX$_3$CR1:GFP* mouse using the single GRIN lens (myeloid *CX$_3$CR1$^+$* cells - green; vasculature labeled by Qdots - red). The maximum field of view is circular, with 280 μm diameter. **f** The PSF was measured on 100 nm beads ($\lambda_{em} = 605$ nm, $\lambda_{exc} = 850$ nm) in agarose, using the single GRIN lens. No significant wave-front distortions affecting the PSF are observed. **g** Qdots are used to estimate PSF in marrow tissue. They reveal slight resolution deterioration with increasing imaging depth. **h** 2D fluorescence images of Qdots-labeled femoral vasculature, 35 days post-surgery, at various z-positions between the surface of the single GRIN lens and 204 μm tissue depth. They reveal fine vascularization in the upper layers and a large blood vessel (~100 μm diameter) with emerging branches in the deep marrow. **i** 2D fluorescence images of femoral vasculature acquired at various depths and time points post-surgery, using the triplet GRIN lens. The tissue at the contact surface with the window is characterized by de novo micro-vascularization, i.e., granulation tissue. Its thickness varies between individuals and decreases over time after implantation. Scale bars = 100 μm

both with a diameter of 350 μm. The first design (Fig. 1d, upper panel) consists of one single GRIN lens of 4.49 mm in length, a numerical aperture (NA) of 0.5 at the object side and a field of view of 280 μm in diameter (Fig. 1e). The lens is permanently and stably glued into the tubing, thereby sealing it to maintain sterility within the marrow cavity and ensuring fixed positioning of the optical path. The second design (Fig. 1d, lower panel) consists of a system of three GRIN lenses with NA 0.42 at the object side,

5.99 mm in length and a field of view of 150 μm diameter. This GRIN system requires a 170 μm thick sapphire window for optimal optical performance, which is attached to the tubing and seals the implant. Thus, this design allows flexible replacement of the GRIN lens system for applications requiring different optical properties. A 45° prism can be glued at the end of the GRIN lens to achieve a side view of the marrow tissue and access additional tissue areas.

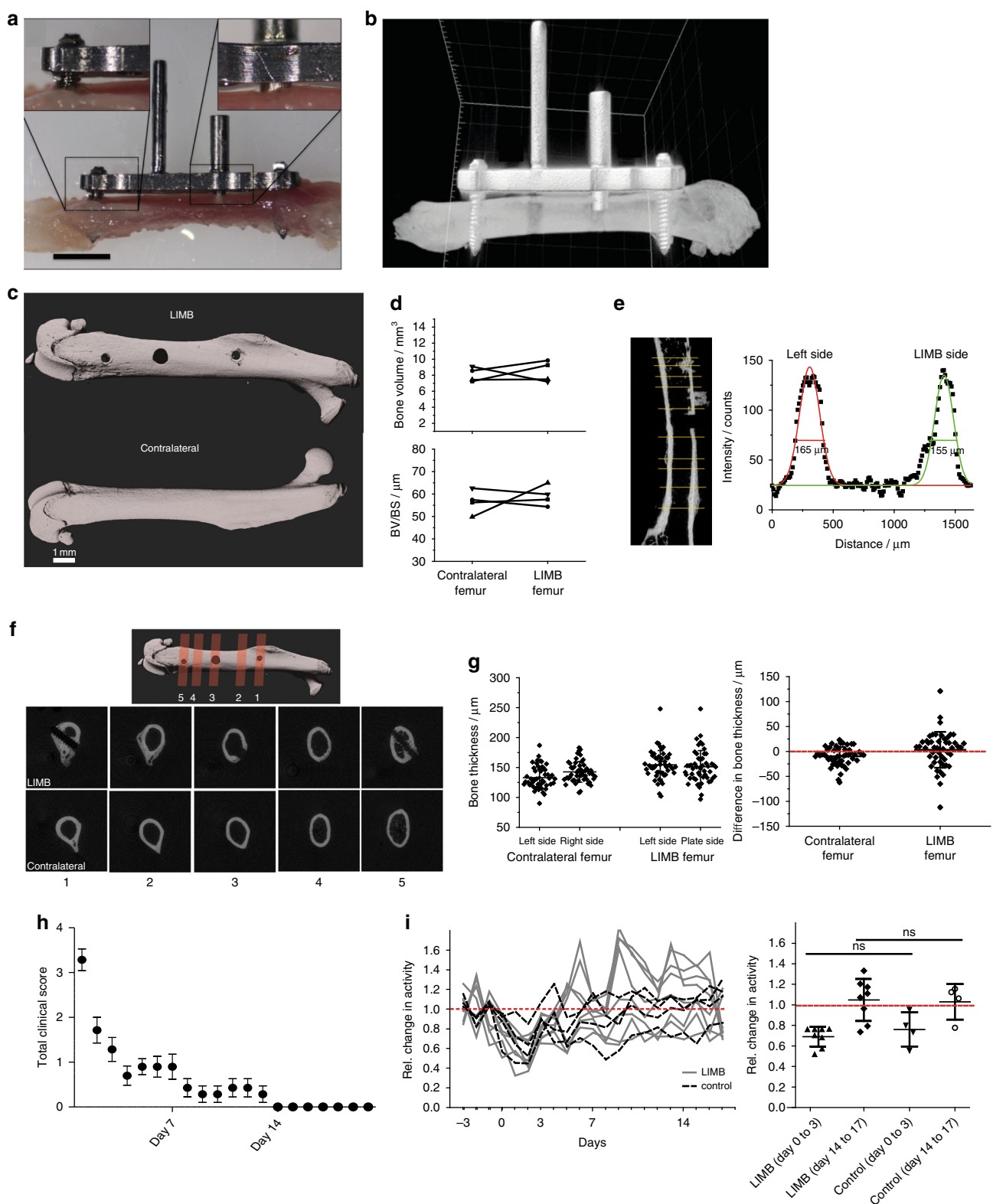

The LIMB implant is surgically fixated onto the right femur of mice (Supplementary Figs. 2b, 3) to intravitally visualize tissue dynamics by multi-photon microscopy. The use of multi-photon excitation permits imaging deep into tissue. The implant is completed by a titanium-alloy reference plate (Fig. 1b, Supplementary Figs. 1, 2a) mounted onto the positioner and the endoscope tubing, which allows stable, reproducible positioning of the mice under the microscope. This reference plate couples with a custom-built adapter used to align the optical axes of the microscope objective lens (20×, NA 0.45) and of the GRIN lens, thereby transferring the focus from the microscope objective through the GRIN lens into the marrow tissue (Supplementary Figs. 2b, 3h).

To characterize the optical properties of LIMB, we first determined the point spread function (PSF) of the GRIN lens systems using 100 nm fluorescent beads (emission at 605 nm, excitation at 850 nm) in agarose. Using the one-lens GRIN design, the spatial resolution was $0.8 \pm 0.1$ μm (s.d.) laterally and $5.2 \pm 0.5$ μm axially, corresponding to theoretical values for NA 0.5 ($n = 23$ beads). The three-dimensional PSF was not distorted by optical aberrations over the entire field of view (Fig. 1f). Using the three-lens GRIN system with the sapphire glass window, we measured a spatial resolution of $0.9 \pm 0.2$ μm laterally and $8.0 \pm 1.1$ μm axially (Fig. 1g), also corresponding to the theoretical values for NA 0.42 ($n = 18$ beads).

Within the bone marrow, the spatial resolution was evaluated using quantum dots 655 (Qdots). While most of these nanoparticles remained within the blood vessels, some Qdots entered the parenchyma due to the fenestrated sinusoids of the bone marrow and were observed as single fluorescent spheres of sub-resolution sizes. Typically, resolution values of $1.0 \pm 0.1$ μm laterally and $7.4 \pm 1.6$ μm axially were achieved within the imaging volume between 40 and 70 μm distance from the surface of the GRIN lens (at each depth, $n = 8$ Qdots).

The maximum extent of the imaging volume within the bone marrow using the one-lens GRIN system started from the endoscope surface at 0 μm and reached down to 204 μm within the bone marrow (Fig. 1h). It included regions with superficial small sinusoids and arterioles, as well as deeper-laying large blood vessels including the central sinus. We reliably achieved a signal-to-noise ratio of at least 5, independent of inter-individual variance and of the imaging time point after surgery.

At early time points post-surgery, we typically visualized granulation tissue, characterized by a tight network of small blood vessels within a large area in the imaging volume (~30–85 μm in depth). Over time, this granulation tissue retracted to the area next to the GRIN optics and was replaced by normal-appearing bone marrow vasculature, resembling the vasculature as observed by calvarial and tibial imaging preparations. At late time points post-surgery, from 28 days onwards, we observed complete resolution of this granulation layer (Fig. 1i).

**Local and overall tissue recovery after implantation.** First, we determined to what extent the surgical procedure and the presence of the implant itself affected the mice and impacted on bone physiology. As shown in Fig. 2a, b and Supplementary Movie 1, we confirmed that the low contact plate design does not compress the bone, instead, a gap of at least 100 μm remained between bone surface and plate. After removal of the plate (day 7 post-implantation), reconstructed μCT images of the bone surface appeared smooth in the area where the plate had been, comparable to the surface of contralateral bones (Fig. 2c). Furthermore, no differences in bone volume, ratio of bone volume to bone surface (BV/BS), bone shape or thickness were found between the LIMB-implanted and contralateral femurs (Fig. 2d–g). In order to determine the impact of the surgery on the mice, we clinically scored the animals immediately after LIMB implantation, taking parameters such as weight, general appearance, and vitality into account. The data recorded from 79 mice indicate that the animals reached a low level of burden (as reflected in a score of 1) by post-surgical day 3, and normal values for all parameters were reached within 14 days after surgery (Fig. 2h). In addition, the general activity of mice that received the LIMB implant was recorded, starting 3 days pre-surgery until 18 days post-surgery (Fig. 2i, left panel). By transponder-based tracking, we recorded the movement of each mouse, thereby generating individual motility profiles of LIMB-implanted and co-housed mice, which had undergone sham treatment (without implantation). The mean activity levels dropped to a similar extent in both implanted and sham-treated groups within 3 days post-surgery, when compared to pre-surgical levels. By day 14 after surgery, at a time point when clinical scores were normal again (Fig. 2h), the activity of the mice in both groups also returned to pre-surgical levels (Fig. 2i, right panel). Interestingly, at both early and late time points post-surgery, there were no significant differences between sham-treated and LIMB-implanted mice. In order to analyze the effects of the surgical intervention and of the implant on the gait of the animals, the open-field behavior of the mice was recorded (Supplementary Movie 2) at day 2, 6, 9, 13, and 21 post-surgery. No severe gait abnormalities were observed at any time ($n = 79$).

In order to assess the effects of the LIMB implant on the bone with respect to post-surgical inflammation and bone formation adjacent to the LIMB microendoscope, we performed immuno-fluorescence and histochemical analysis (Fig. 3a–c). The most striking changes within the marrow of the implant-bearing

**Fig. 2** Effect of LIMB implantation on bone tissue, general health, and activity of the mice. **a** Picture of an explanted femur including the LIMB implant shows the fixation plate bridging between the angle-stable screws, thereby preventing direct contact to the bone surface/periosteum. Scale bar = 2 mm. **b** 3D reconstructed μCT images of a femur bearing the LIMB implant confirm no direct contact between bone and fixation plate. Shadows below the positioner are beam hardening artifacts of the high attenuation titanium alloy. **c** 3D reconstructed μCT images of an intact femur (lower panel) and a femur after removal of the LIMB implant (upper panel). Bone surface under the fixation plate 7 days after implantation appears similar to the bone surface of the intact femur. Bone growth is observed only around the bicortical screws. **d** Total bone volume (upper panel) and BV/BS (lower panel) of LIMB and contralateral femur, respectively ($n = 4$ mice). **e** Longitudinal cross-section through a μCT reconstruction of a femur after implant removal (left panel). Yellow lines represent the positions chosen to measure the bone thickness. For each position, the intensity profile was approximated with two Gaussian curves as indicated in the left graph. **f** 3D reconstruction of μCT data showing the planes of transverse cross-sections displayed in the lower two panels. At day 21 after implantation, calcified bone forms around the bicortical screws, but not around the endoscope tubing. Cross-section μCT images at the sites of the screws show enough space for the bone marrow tissue to connect the diaphysis with the metaphysis. **g** No differences in bone thickness between contralateral and LIMB femurs are measured ($n = 5$ mice). **h** Total clinical score over 3 weeks post-surgery based on behavior and appearance of the individual mice ($n = 79$ mice). **i** Physical activity of LIMB implanted mice, pre-surgery and post-surgery, and of co-housed control mice. The LIMB-implanted mice reach their pre-surgical activity level within the same time as sham-treated controls (right graph) (two independent experiments; $n = 8$

femurs were an accumulation of CD45$^+$ cells, and enhanced expression of the extracellular matrix (ECM) component laminin in tissue regions around the implant, peaking at day 3 post-surgery. Some of the cells with hematopoietic morphology were Sca-1$^+$ckit$^+$ and may represent progenitors (Supplementary Fig. 4a). We also detected CD45$^-$Sca-1$^+$ cells in elongated structures surrounded by laminin, likely representing arterioles

sprouting into the injured area (Supplementary Fig. 4b). These changes occurred around the sides of the implanted tube and in front of the sapphire glass window, reaching a depth of up to 400 µm. By day 7, the inflamed area as indicated by laminin, Sca-1 and CD45 shrunk to a thickness of ~100 µm. The tissue structure further normalized at day 14 and by day 28, the area in front of the imaging window typically consisted of laminin$^+$ vessels

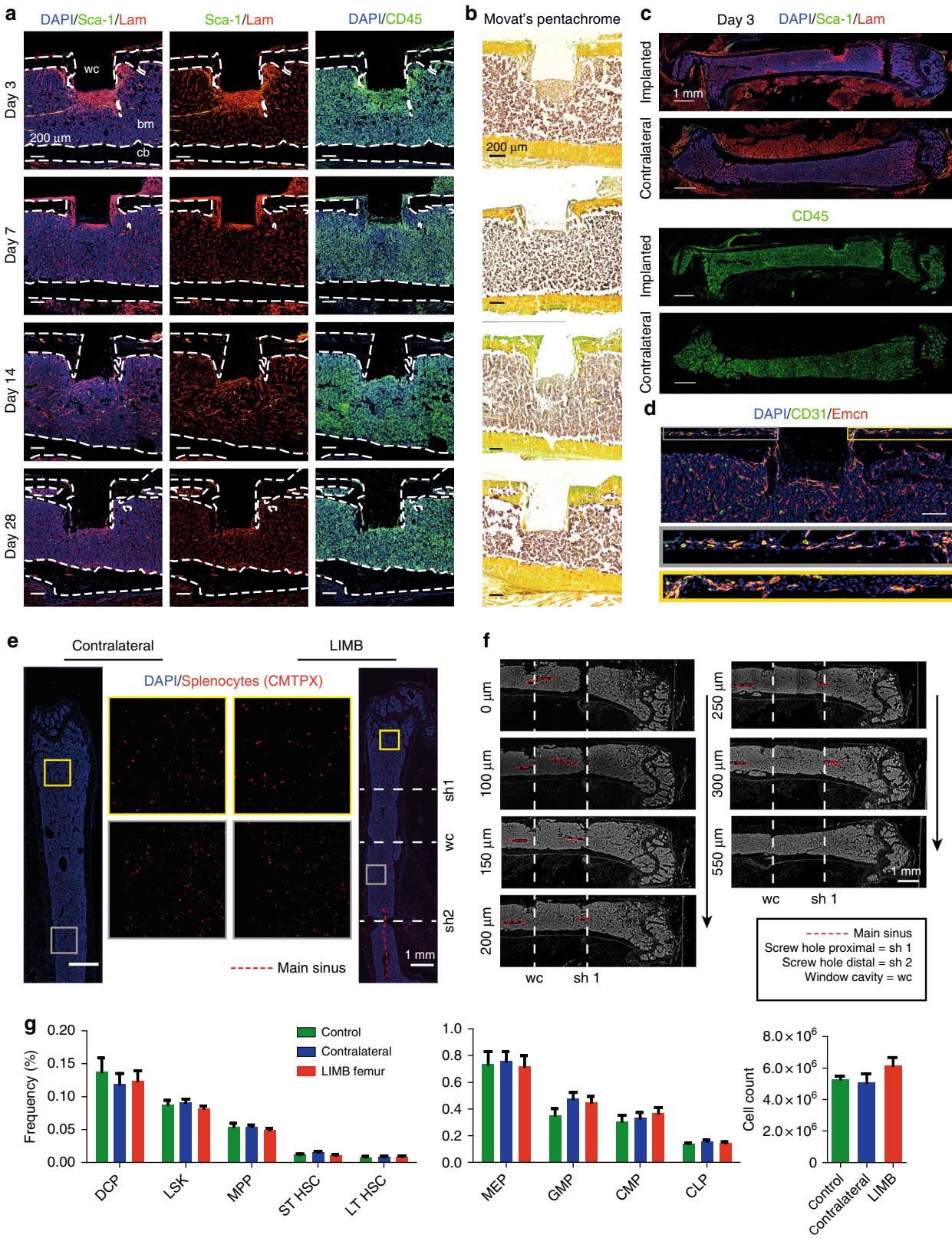

surrounded by CD45[+] hematopoietic parenchyma, in a pattern resembling the other regions of the bone marrow and the contralateral (non-implant-bearing) femur. Movat's pentachrome staining showed that, although in some mice bone formation occurred lateral to the implanted tube emanating from the cortex, no new bone formed in front of the imaging window. Besides these changes in the tissue occurring adjacent to the implant and screws, the overall structure of the bone marrow was unaltered with respect to the overall distribution of leukocytes, ECM components, and vessels when compared to the contralateral bone (Fig. 3c). At the site of the periosteum, vascularization was detected (Fig. 3d). In order to test whether the implant impairs blood flow throughout the bone, we adoptively transferred CMTPX-labeled splenocytes via the tail vein, harvested the bones 4 h later and analyzed the distribution of CMTPX[+] cells by fluorescence histology. In histological sections, CMTPX[+] cells were distributed throughout the marrow in both the contralateral and the LIMB-implanted femurs (Fig. 3e). Importantly, engraftment occurred on both sides of the bicortical screws. Consistent with this observation, longitudinal serial sections showed clearly that the bicortical screw did not completely separate the bone marrow and did not impair the blood flow in any part of the marrow. Additionally, we performed μCT of implanted bones, which also showed clearly that the bone marrow was not completely separated by the bicortical screws (Fig. 3f and Supplementary Movie 1). In order to verify whether hematopoiesis within the bone marrow was altered, we performed flow cytometry on bone marrow cells of LIMB-implanted mice, and quantified the major hematopoietic progenitor populations. No significant differences in the frequencies or total cell counts of the analyzed populations were observed compared to bone marrow from contralateral femurs or bone marrow from non-treated mice (Fig. 3g and Supplementary Fig. 5).

**B lymphocyte motility in femur, calvarium, and tibia.** In order to observe the dynamics of B-lineage cells within the femoral marrow, we chose CD19:tdRFP mice as recipients of the LIMB implant. In these mice, Cre recombinase activity in CD19-expressing cells leads to the removal of a STOP cassette and expression of tdRFP in the B cell lineage, starting in the late pro-B cell stage. The vasculature was labeled using Qdots. Mice were imaged starting at day 7 (Fig. 4a; Supplementary Movies 3, 5–7), up to maximally day 115 post-surgery (Supplementary Movie 12).

Using LIMB in CD19:tdRFP mice, under homeostatic conditions, i.e., at day 60 or 90 post-surgery, we found that tdRFP[+] cells of various volumes displayed different motility patterns. We distinguished the subtypes of B lymphocytes based on their size,

and tracked them to assess their velocity and displacement rate. A maximum diameter of 10 μm corresponds to a spherical volume of ~500 μm[3], therefore these cells presumably represent B cells, displaying a high degree of directed motility, in line with previous reports[5]. Larger tdRFP[+] cells in this system represent rather static plasma cells, consistent with the concept of their residence in bone marrow niches (Fig. 3c) and with our previous work[3]. In order to compare the motility of B-lineage cells in the femur to other bone marrow compartments, we performed intravital imaging of calvaria and tibia of CD19:tdRFP mice. Notably, we found no significant differences in the motility of bone marrow B and plasma cells when comparing femur (LIMB), calvarium, and tibia (Supplementary Movies 4, 8, 9; Fig. 4b–e).

**Long-term stability of the LIMB implant.** In order to test the positional stability of the LIMB implant, the microendoscope was implanted in mice ubiquitously expressing photoactivatable green fluorescent protein (paGFP). Prior to photoactivation, paGFP is non-fluorescent upon 940 nm illumination. Photoactivation at 840 nm leads to 100-fold increase in green fluorescence upon excitation at 940 nm[28]. PaGFP mice were injected with Qdots to label the vasculature and were imaged before and after photo-activation (Fig. 5a, two upper rows). The photoactivation area was either 75 × 75 or 100 × 100 μm[3] positioned within the center of the field of view (Supplementary Fig. 6). We analyzed paGFP fluorescence by repeated imaging every 6–12 h (Fig. 5a, third to last row) and detected paGFP fluorescence at the same area at least 36 h after photoactivation. Thus, we confirmed the high positioning stability of the LIMB implant with respect to the implantation/imaging site in the bone. The fluorescence signal of paGFP decreases over the time span of hours/days due to the emigration of hematopoietic cells out of the photoactivated region (Supplementary Movies 9, 10) and due to protein turnover, causing fluorescent paGFP to be replaced by newly synthesized, non-fluorescent paGFP. The imaging data acquired in paGFP mice reveal that while the field of view labeled by photoactivation remains stable, morphological changes of the vascular system do not cease after day 28 post-surgery and seem to depend on the vessel diameter (Fig. 5a). In order to exclude that such changes are caused by the LIMB implant itself, we performed longitudinal imaging in the calvarium of paGFP mice, over a volume of 500 × 500 × 66 μm[3]. We were able to repeatedly detect paGFP fluorescent areas in the calvarial marrow islets for up to 72 h after photoactivation, and observed paGFP cells leaving this area. Similarly to LIMB, we also detected drastic changes in the marrow vasculature, most prominent in smaller vessels (Fig. 5b; Supplementary Fig. 7).

**Fig. 3** The bone marrow within the imaging volume reaches steady-state comparable to homeostasis 28 days after LIMB implantation. **a** Immunofluorescence analysis of bone sections after removal of the LIMB implant over the time course of 4 weeks. ECM formation was identified by the marker Laminin (Lam). Stem-cell antigen 1 (Sca-1) is highly expressed in arterioles. The leukocyte marker CD45 indicates localization of inflammatory cells adjacent to the window cavity (wc). Lam is highly expressed around the implant during the first week and completely normalizes after 2–4 weeks. CD45[+] cell accumulations are found during the first weeks in proximity to the wc. **b** Movat's pentachrome stain detects connective tissues and reveals remodeling of bone primarily on the periosteal interface near the fixation plate. **a**, **b** Images are representative for 3–5 mice per time point post-surgery. **c** Overview immunofluorescence images of the femoral bones from an individual mouse 3 days post-surgery. Note the specific reaction to the implant-bone marrow interfaces indicated by accumulations of CD45[+] cells, and Lam[+] and Sca1[+] arteries (yellow). bm bone marrow, cb cortical bone. **d** Immunofluorescence image of the region around the endoscope tubing in a LIMB-implanted femur 7 days post-surgery, including the bone cortex and periosteum under the plate. The presence of various blood vessel subsets indicated by CD31 and Emcn demonstrates intact blood supply to the periosteum and to the bone. Scale bar = 100 μm. **e** Blood supply is intact throughout the marrow cavity, indicated by CMTPX-labeled splenocytes, which localize in the bone marrow 4 h after transplantation in both contralateral and LIMB-implanted femurs at 42 days post-surgery (n = 3 mice). **f** Histological DAPI stain (gray) shows intact tissue structure, with no separation of the bone marrow and the vasculature by the screws or endoscope tubing. **g** Flow cytometry analysis of femurs with the LIMB implant, their contralateral femurs and femurs of control mice. Similar frequencies and cell counts of various cell populations shows no effect of the LIMB implant on bone marrow cell composition (n = 8 LIMB-implanted mice, n = 8 controls, two independent experiments). Error bars represent s.e.m. values. Statistical analysis was performed using t-test

**Vascular plasticity during healing phase and homeostasis.** Using LIMB, we observed drastic changes in the vascular morphology within the femoral marrow after implantation, both during the healing phase and after reaching steady-state. These changes were not caused by instability of the microendoscope or by inaccuracy during repeated selection of the imaging volume, as demonstrated by the results of our photoactivation experiments.

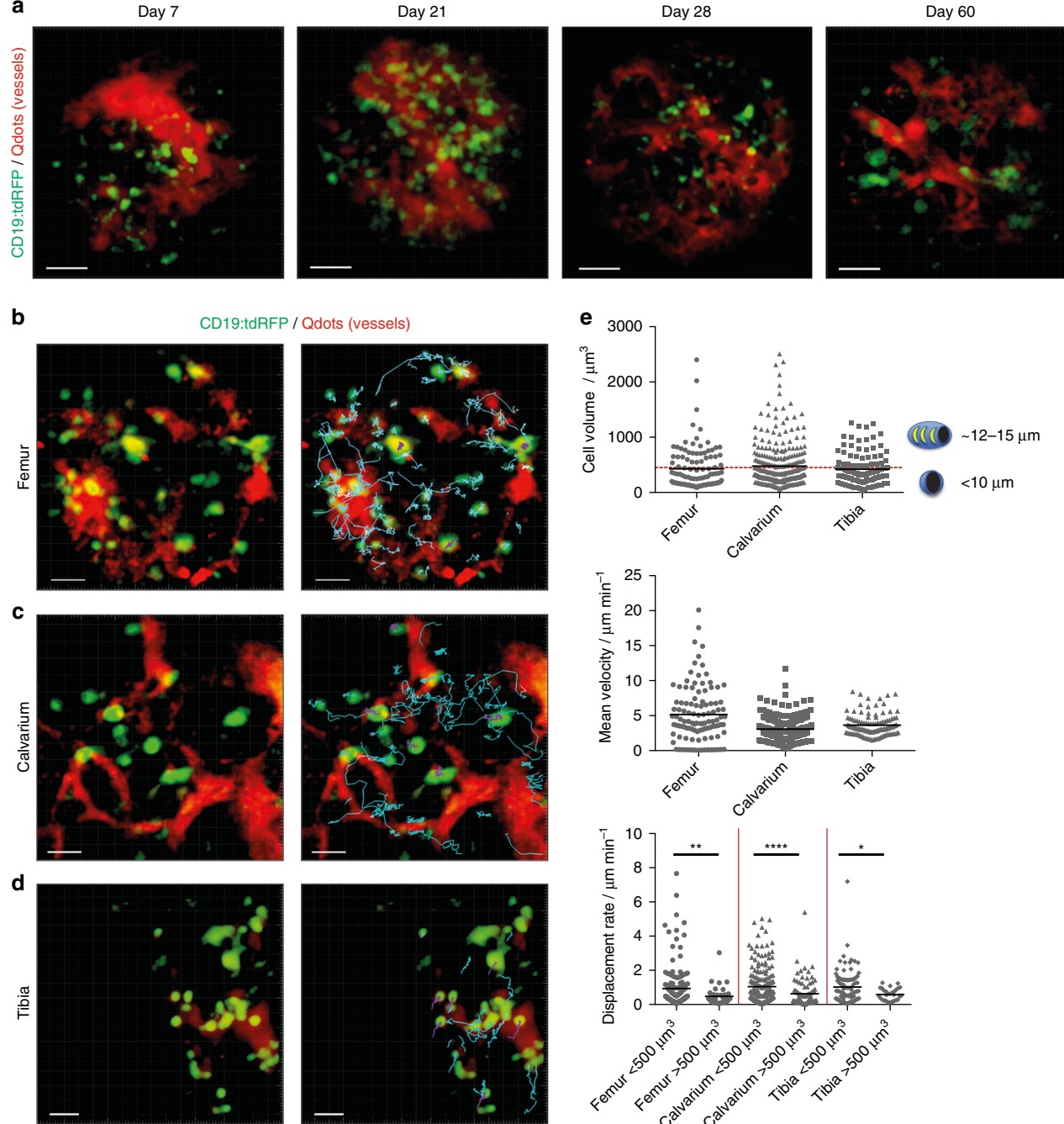

**Fig. 4** Immune cell dynamics in different bone types show comparable motility patterns. **a** 3D fluorescence images acquired by LIMB in the femoral marrow of a *CD19:tdRFP* mouse at day 7, day 14, day 28, and day 60 after surgery. Mature B lymphocytes express tdRFP and are displayed in green, whereas the vasculature was labeled with Qdots and is displayed in red. Scale bar = 50 µm. All images are snapshots of 45 min movies, with images acquired every 30 s. The movies are provided as Supplementary Material. **b** Time-lapse 3D fluorescence image acquired by LIMB in the bone marrow of a *CD19:tdRFP* mouse at day 90 post-surgery. The tracks of the B lymphocytes smaller than 500 µm³ (defined as B cells with a maximum diameter of 10 µm) are shown in cyan, whereas those with a volume larger than 500 µm³ (defined as plasma cells) are shown in violet. Scale bar = 30 µm. **c** Similar to **b**, time lapse 3D image of the bone marrow of a *CD19:tdRFP* mouse with corresponding tracks of B and plasma cells in the calvarium and **d** the tibia. Scale bar = 30 µm. Representative movies with cell motility tracks for LIMB, calvarial, and tibial imaging are provided as Supplementary Material. **e** Quantification of cell volumes, mean velocities, and displacement rates of B lymphocytes from movies acquired by LIMB (*n* = 4 mice), within the calvarial bone (*n* = 3 mice), and the tibia (*n* = 2 mice). Similar cell subset frequencies and mean velocities of B lymphocytes were measured by LIMB in the femoral bone marrow, by calvarial imaging as well as by tibial imaging. We statistically analyzed the data in **e** using *t*-test (*p < 0.05; **p < 0.01; ***p < 0.001)

As expected, we observed dynamic blood vessel re-organization between day 7 and 21 post-surgery. Interestingly, even between day 27 and 60, a time span in which the tissue composition in front of the microendoscope is comparable to non-implanted bone marrow (Figs. 2, 3), the vasculature continued to change massively (Fig. 6a). Starting from day 27 post-surgery, with a time

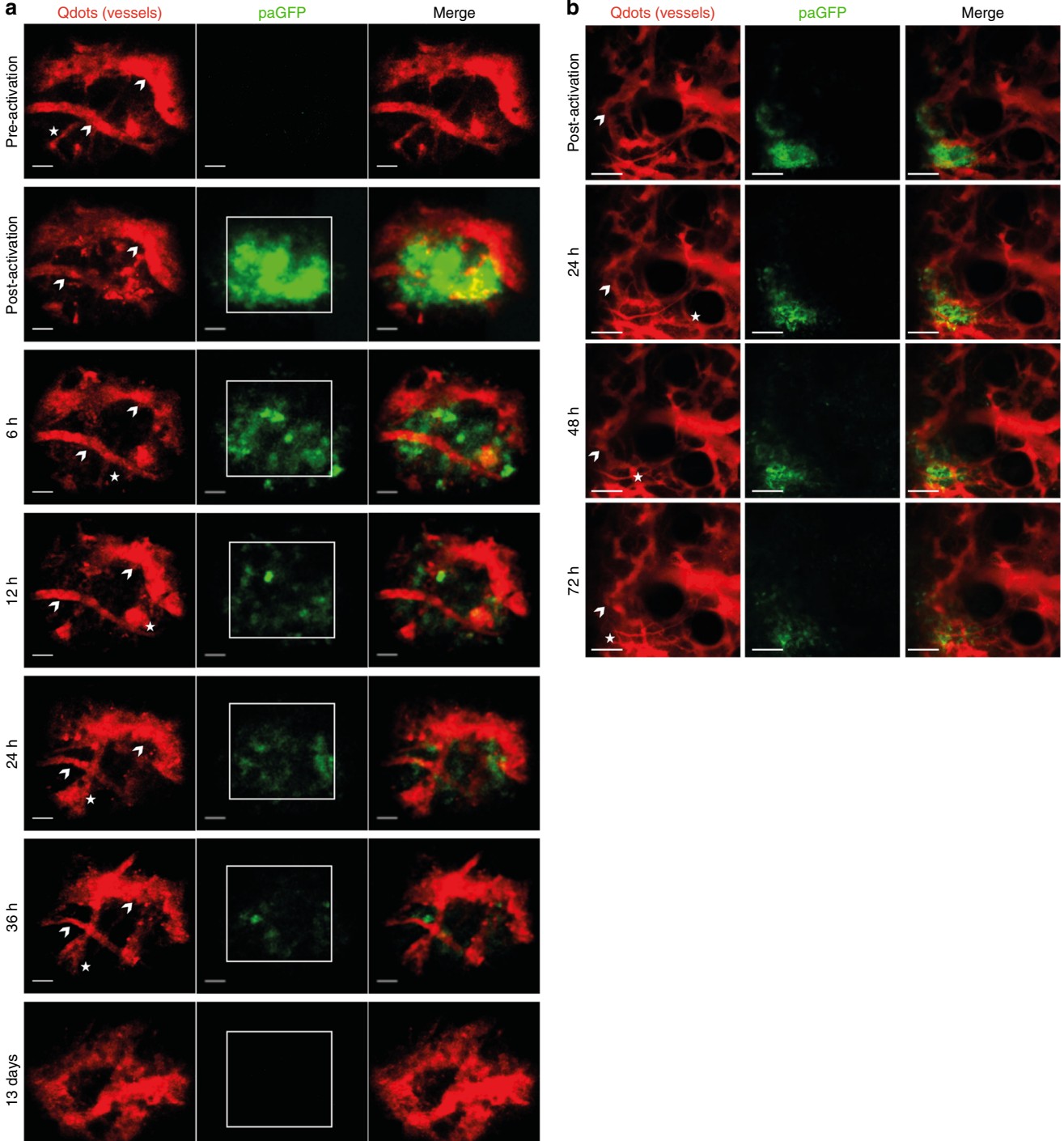

**Fig. 5** Imaging of locally activated paGFP in murine deep femoral marrow reveals high positioning stability of the LIMB implant. **a** paGFP mice were implanted with a LIMB microendoscope (n = 3 mice). After 35 days, the mice were injected intravenously with Qdots to label the vasculature. Photoactivation of paGFP was performed at a wavelength of 840 nm in a 75 × 75 × 30 μm³ square area in the center of the field of view. Additional injections of Qdots were given before each recording. We performed the described photoactivation experiments repeatedly, up to three times in the same animal at day 27, 35, and 56 post-surgery, during homeostasis with similar results. Blood vessels which could be observed over the whole period of 36 h are indicated by arrowheads, whereas those that appear or disappear within this time period are labeled by asterisks. Scale bar = 30 μm. **b** Similarly to **a**, photoactivation of a 150 × 150 × 9 μm³ region within the 500 × 500 × 66 μm³ field of view in a paGFP mouse with a permanent calvarial imaging window let us easily identify the photoactivated area. The paGFP fluorescence could be visualized over several imaging sessions. Scale bar = 100 μm. During these time windows we observed changes of the vasculature in both, the deep femoral marrow and bone marrow islets of the calvarium

resolution of 7 days between imaging sessions, the pattern of large blood vessels (>50 µm) changed between consecutive time points, making their use as tissue landmarks impossible (Fig. 6b). Only by further increasing the time resolution in LIMB, during the homeostatic phase, i.e., shortening the interval between imaging sessions to every 6–24 h (Fig. 6c), were the vessels stable enough

to be used for orientation. We found that blood vessels with diameters in the range of 5–15 µm had the highest degree of volume change within the time span of 24 h (Fig. 6d, Supplementary Movie 13). Within the same period, large blood vessels (>35 µm) changed rather slowly, whereas middle-sized blood vessels with a diameter between 15–35 µm displayed an

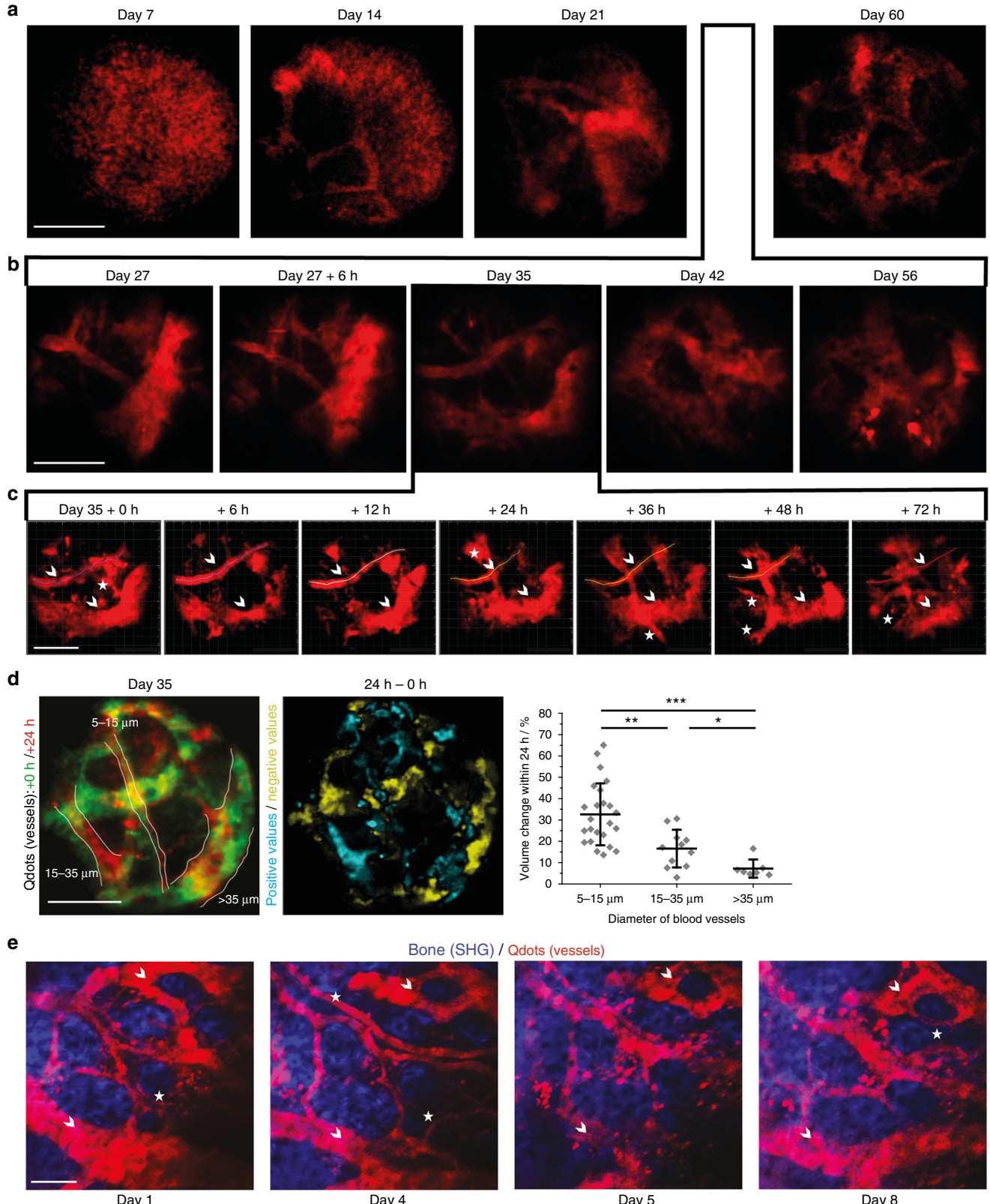

intermediate degree of remodeling. Notably, we observed a comparable remodeling in the vascular compartment of the marrow islets of the calvarium, over a time course of 8 days (Fig. 6e). As the surgery for longitudinal calvarial imaging does not disrupt the integrity of the bone, we can rule out that the vascular plasticity was an effect caused by the femoral implant and confirm the physiological nature of this vascular remodeling. However, due to the fact that calvarial marrow islets are rather small as compared to the cortical bone areas, the dramatic vascular plasticity remained underestimated until now. Only by using LIMB, we were able to observe and quantify these changes.

**Possible mechanisms of femoral vascular plasticity**. In order to investigate whether vascular remodeling was the result of active angiogenesis related to de novo bone formation, we performed immunofluorescence analysis of bone sections (Fig. 7). A recent publication described CD31[hi] Endomucin (Emcn)[hi] (type H) bone marrow endothelium to be involved in osteogenesis-related angiogenesis in the bone marrow[7]. In line with previous reports[29], we found type H vessels to be abundant at the growth plate of young mice, but only few type H vessels were present in aged mice (Supplementary Fig. 8a). In contrast, type H vessels close to the bone cortex were found at all ages (Supplementary Fig. 8a), consistent with age-independent bone remodeling at this site. We examined the bone marrow for the presence of type H vessels following implantation (Fig. 7a) and detected them during the early healing phase (days 3–14) in tissue areas next to the microendoscope. At day 28 post-surgery, only few type H vessels were found in proximity of the microendoscope and their shape and abundance were comparable to the vessels present at bone–bone marrow interfaces. Together, this indicates a state of homeostasis, consistent with the previously observed time course of regeneration (Fig. 3).

In line with the vascular remodeling within the bone marrow, we observed marked changes in the stromal network in *Prx-1:YFP* mice (Supplementary Fig. 9). During homeostasis, the stromal network remodeling continues, similar to the vascular reorganization. Repeated imaging experiments in *paGFP* mice every 6–12 h after photoactivation, over a total time of 36 h (Fig. 5), revealed fluorescent cells, presumably stromal cells, persisting over the whole time span of 36 h, as well as highly motile fluorescent cells, presumably hematopoietic cells, leaving the photoactivated volume. Over the course of these 36 h the vasculature continuously changed its shape. In order to assess whether the remodeling process involves proliferation of the endothelial cells, we administered the thymidine analog EdU to label newly synthesized DNA[3] to mice that had received a LIMB implant either 3 or 42 days earlier. EdU labeled a similar fraction and

comparable pattern of hematopoietic cells distributed over the whole bone marrow in both cohorts ($n = 3$ mice in each cohort). We did not detect an overt proliferation of endothelial cells in the tissue surrounding the implant (Fig. 7b). Similarly, staining for the proliferation marker Ki67 did not show any accumulation of proliferating endothelial cells at the implant site, at any time point (Supplementary Fig. 8b). Taken together, these data support the hypothesis that the observed dynamics are the result of passive displacement of the vessels, probably caused by dynamics and cell proliferation in the surrounding tissue (Fig. 7c; Supplementary Movies 10, 11), rather than active proliferation in the vascular compartment. The exact kinetics and mechanisms of the vascular and stromal remodeling during the steady-state will be subject of further studies, as they impact on many processes taking place in the bone marrow.

## Discussion

While intravital imaging of the bone marrow has previously been performed[3, 11–18, 22], longitudinal multi-photon imaging in the deep marrow of long bones over the time course of months was not feasible. Longitudinal microscopic and microendoscopic imaging of the CNS[20, 21], retina[30], or lymph nodes transplanted into the ear of mice[31] has been reported, and blood vessels have been used as anatomical reference points, allowing the recognition and imaging of the same regions within the tissue.

The bone marrow contains quiescent and activated hematopoietic stem cells in dedicated perivascular tissue niches[6, 29], and is the tissue in which most leukocytes complete their development. The bone marrow actively participates in immune reactions[32, 33] and is an important site for the maintenance of immunological memory[3, 10, 34, 35]. The interaction of hematopoietic cells with stromal cells plays a crucial role in these processes: subtypes of vascular cells provide niches for the maintenance of hematopoietic stem cell quiescence[6], and stromal cells in the bone marrow provide cytokines, which support the survival and differentiation of developing hematopoietic cells. Endothelial cells mediate leukocyte trafficking between the circulation and the bone marrow parenchyma[6], guiding immune memory cells back into the bone marrow together with stromal cells, which provide chemotactic cues and anchor the memory cells in their niches[36]. Moreover, the role of the immune, vascular and bone compartments during distinct phases of bone regeneration still remain elusive. Hence, there is a clear need for longitudinal imaging technologies to allow the investigation of dynamic processes at the cellular and sub-cellular level in this organ.

The GRIN lens microendoscopic implant presented in this work allows for the first time imaging of dynamic cellular

---

**Fig. 6** LIMB approach reveals kinetics of vascular remodeling during bone healing and homeostasis on time scales from hours to months. *C57/B6J* mice received the LIMB implant and were injected intravenously with Qdots (red) prior to each LIMB imaging session to label the vasculature. Vessels were three-dimensionally imaged at increasing time resolution over the course of **a** weeks **b** days and **c** several hours. In line with our previous observations, we noted prominent changes in the vasculature, which continued over the whole monitoring time period, even after homeostasis is reached ($n = 5$ mice, two independent experiments, scale bar = 50 μm). Small vessels continuously appear and disappear, larger vessels change their position and shape. The trace of such a larger vessel is displayed at all time points as a line in **c**. Blood vessels which can be used as landmarks are labeled by arrowheads and those that completely disappear within days are labeled by asterisks. **d** Overlap of the 3D projections of blood vessels in a mouse 35 days post-surgery (+0 h, green) and 24 h later (+24 h, red). A differential image between the two 3D images was generated. Blood vessel volume change was calculated by dividing the fraction of the volume difference between +24 h and 0 h (cyan areas in the middle panel indicate positive values, i.e., appearance of blood vessels; yellow areas indicate negative values, i.e., disappearance of blood vessels) by the total volume of the blood vessel at +24 h (delineated by white lines in the left panel) to obtain a normalized parameter of vessel volume change. The normalized volume changes (right panel) are dependent on the blood vessel diameter, with small vessels remodeling more rapidly than large vessels ($n = 6$ mice, scale bar = 100 μm). **e** Similar to the observations in the deep femoral marrow, repeated imaging of blood vessels in calvarial bone and bone marrow also showed remodeling of the vasculature ($n = 3$ mice, two independent experiments). Scale bar = 100 μm. Error bars represent s.d. values. Statistical analysis in **d** was performed using an ANOVA test (*$p < 0.05$; **$p < 0.01$; ***$p < 0.001$)

processes in the deep femoral marrow, repeatedly, over a time period of up to several months. The central marrow represents an area, which has been inaccessible to intravital multi-photon imaging until now; our technique permits imaging of this tissue for the unprecedented duration of 4 months within the same bone marrow region. Importantly, this time period covers not

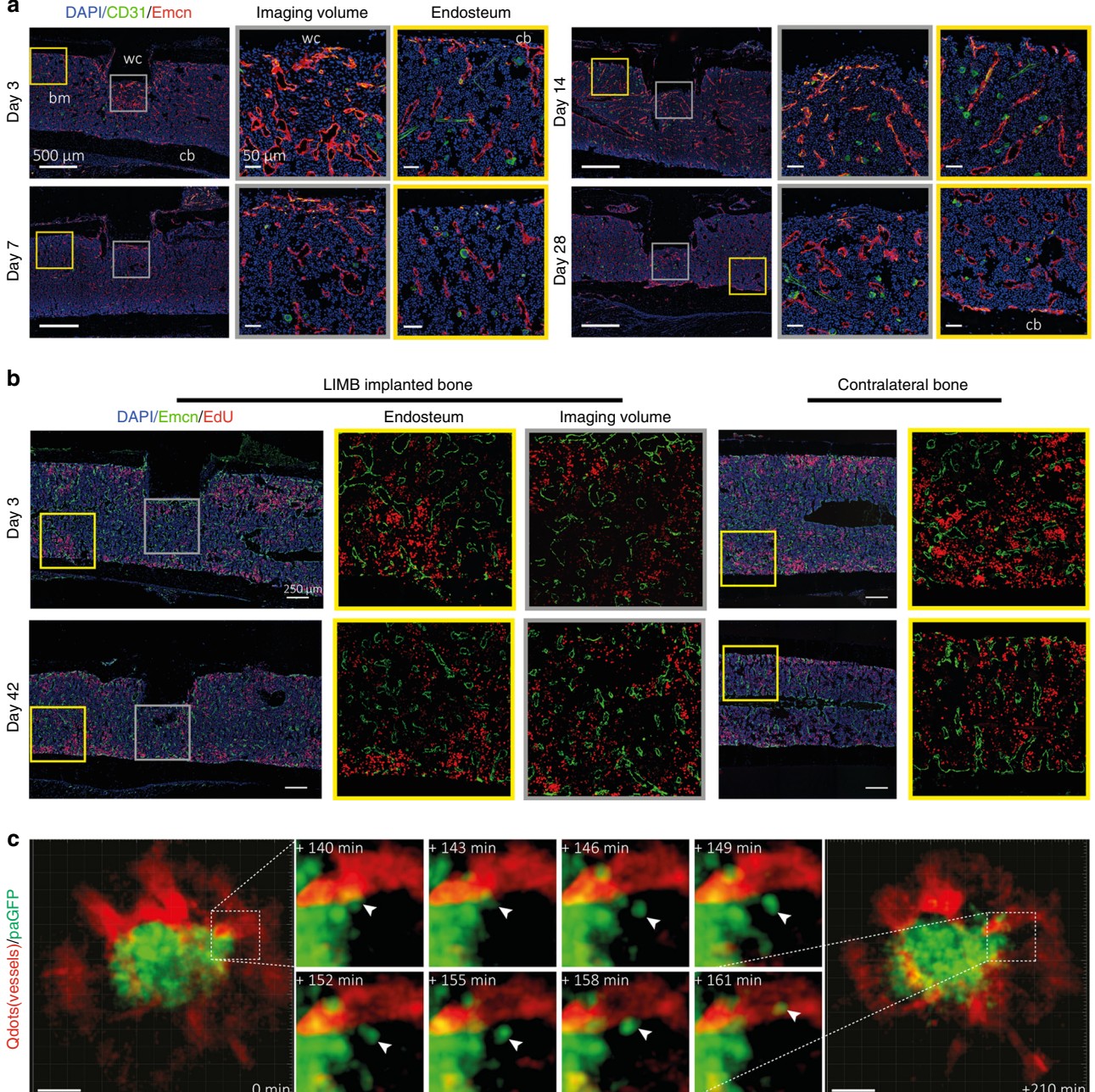

**Fig. 7** LIMB and immunofluorescence analysis indicate possible mechanisms of vascular morphological changes deep in the femoral bone marrow, during regeneration, and in steady-state homeostasis. **a** Immunofluorescence analysis shows that type H vessels, characterized by CD31$^{hi}$Emcn$^{hi}$-expressing endothelial cells, are induced and present around the implant at day 3 after LIMB implantation. Their presence may vary individually but normalizes within 28 days post-surgery. Sinusoidal and type H vessel morphology adjacent to the wc is irregular in the first week and completely reorganizes to an appearance comparable to vessels found at endosteal areas distant from the injury site ($n = 3$ mice). bm bone marrow, cb cortical bone. Scale bar = 500 µm (left panels). **b** Immunofluorescence analysis after EdU pulse-chase experiments indicates similar EdU-uptake in the bone marrow of LIMB-implanted femurs and contralateral bones. Proliferating endothelial cells were rarely present at late time points after implantation. This result also supports the conclusion that 28 days after LIMB implantation both the bone and the bone marrow reach homeostasis ($n = 3$ mice in each cohort). **c** 3D fluorescence image ($300 \times 300 \times 66$ µm$^3$, left and right panel) acquired by LIMB 26 days post-surgery, in a *paGFP* mouse with the vasculature labeled by Qdots. Photoactivation was performed within a volume of $100 \times 100 \times 9$ µm$^3$ in the center of the image. The fluorescence image was acquired 2 h post activation. Scale bar = 50 µm. The middle panel shows time-lapse 3D images of the inset from the left panel, indicating that paGFP fluorescent cells outside the initial photoactivation volume are present 3 h after photoactivation and that they fluctuate in number and position within the tissue. Passive displacement of the relatively immobile stromal and vascular compartments by continuous proliferation and movement of hematopoietic cells is a possible mechanism of tissue and vascular re-localization during homeostasis (see Supplementary Movies 10, 11)

only the bone healing phase, but also allows imaging during post-operative steady-state homeostasis, as demonstrated in this work.

As a microendoscopic approach, LIMB allows imaging of areas between 100 and 500 μm deep within bone marrow tissue, which are not accessible to existing intravital microscopy preparations. Whereas the maximum field of view of 280 μm in diameter and the spatial resolution are comparable to those achieved by the established longitudinal conventional intravital multi-photon microscopy, the imaging volume is fixed. This represents both a benefit, since the implant itself functions as a landmark for orientation in tissue, but also a limitation, since covering the relevant tissue areas can be more challenging than with other imaging methods.

Along that line, as we and others have previously shown[3], the deep bone marrow in diaphyseal regions does not show a supra-cellular compartmentalization, in contrast to secondary lymphoid organs, which are divided into zones that primarily host certain immune cell subtypes. In order to address tissue heterogeneity in the bone marrow, we developed various designs of our implant that allow positioning of the GRIN lens at various locations within the bone marrow, based on known anatomical compart-mentalization. Hence, LIMB allows the imaging of diaphyseal or metaphyseal areas as well as peri-cortical vs. deep-marrow regions, which are known to differ in their cellular composi-tion. As we demonstrated by imaging sessile plasma cells (which are present in the bone marrow in frequencies of less than 1% of all hematopoietic cells), LIMB is suitable for the analysis of rare cells, despite the rather small field of view. This kind of analysis is crucial to improve our understanding of both hematopoiesis and immunological memory.

Upon fluorescently marking the lumina of blood vessels as a way to identify the same tissue regions in different imaging ses-sions[21, 31], we unexpectedly found the vasculature to be markedly dynamic during steady-state homeostasis. As compared to other tissues, the bone marrow is characterized by a much higher fre-quency of proliferating cells[3], and an additional level of dynamics is added by the motility of hematopoietic cells. There is a constant egress of cells[5], but cells can also enter the marrow tissue, as in the case of memory T cells or plasma blasts[34, 37]. Currently, our experiments using different strategies in order to label pro-liferating cells indicate that dynamics occurring in the non-vascular compartment may affect the positioning of the vessels rather than proliferation of the endothelial cells. Future studies based on LIMB analysis will show whether the observed dynamics are solely caused by a passive displacement of the vessels induced by proliferation or motility of bone marrow cells, or whether they are the result of an active process such as neovascularization, especially in light of recently published findings, which indicate that angiogenesis in the bone marrow mechanistically differs from other organs[38]. In any case, we expect that our results will have implications on our understanding of bone marrow microenvironment organization. They add complexity to the concept of how various tissue niches within the bone marrow are regulated. In the future we will investigate the impact of stromal network dynamics on the niches, since stromal cells may also contribute to the stability of those niches by acting as a scaffold in the tissue.

Concluding, LIMB is a unique method, which enables us to longitudinally analyze the same tissue areas in the deep bone marrow of individual mice by intravital multi-photon micro-scopy. Our technique will open up new ways for analyzing both long-term processes occurring with slow dynamics as well as short-term processes, characterized by fast cellular dynamics. LIMB can be applied to image any process occurring in the bone marrow, including regenerative processes, such as bone formation

in adaptation or after injury, as well as tumor formation and bone marrow metastasis, to name only a few examples.

## Methods

**Two-photon laser-scanning microscopy.** Multi-photon fluorescence imaging experiments were performed using a specialized laser-scanning microscope based on a commercial scan head (TriMScope II, LaVision BioTec GmbH, Bielefeld, Germany). We used a 20× objective lens (IR-coating, NA 0.45, Olympus, Ham-burg, Germany) combined with GRIN lenses (Fig. 1, GRINTech GmbH, Jena, Germany) to focus the excitation laser radiation into the sample and to collect the emitted fluorescence. Detection of the fluorescence signals was accomplished with photomultiplier tubes in the ranges of $(466 \pm 20)$ nm, $(525 \pm 25)$ nm, $(593 \pm 20)$ nm, and $(655 \pm 20)$ nm. paGFP was photoactivated at 840 nm. Both, activated paGFP and GFP were excited at 940 nm and detected at $(525 \pm 25)$ nm. eYFP (Prx-1:YFP mice) was excited at 940 nm and detected at $(525 \pm 20)$ nm and $(593 \pm 20)$ nm, while tdRFP was excited at 1100 nm and detected at $(593 \pm 20)$ nm. Blood vessels were labeled with Qdots excited at 940 or 1100 nm and detected at $(593 \pm 20)$ nm. In all imaging experiments we used an average maximum laser power of 10 mW to avoid photodamage, at a typical pixel dwell time of 2 μs. The maximum peak photon flux was $10^{29}$ photons cm$^{-2}$ s$^{-1}$, in accordance to the values measured using conventional two-photon microscopy. The acquisition time for an image with a field of view of $150 \times 150$ μm$^2$ and a digital resolution of $497 \times 497$ pixel was 800 ms, as well as for a field of view of $350 \times 350$ μm$^2$ and a digital resolution of $507 \times 507$ pixel. We acquired 70 μm z-stacks (z-step size 6 μm) each 30 s over a total time course of typically 45 min. Calvarial and tibial imaging were performed using the same microscope setup. In contrast to LIMB, we used a 20× water-immersion objective for focusing (IR coated, NA 0.95, Olympus, Hamburg, Germany).

**Data analysis.** Image segmentation and tracking of all cells was performed using segmentation, object-recognition, and tracking plugins in Imaris (Bitplane AG, Zurich, Switzerland). Cell tracks that were present in the field of view for more than 10 recorded time points (i.e., 5 min) were included in the analysis. Statistical analysis of the data was performed using Prism (Graph Pad Software Inc., San Diego, USA).

**Mice.** All mice used were on a C57/Bl6J background. CD19:tdRFP fate mapping mice were generated in our lab by crossing CD19:Cre[28] onto ROSA26:tdRFP[29] mice. Prx-1:Cre mice[30] were crossed onto ROSA26:eYFP mice (Prx-1:YFP). Pho-toactivatable GFP (paGFP) mice, which ubiquitously express paGFP[18], were kindly provided by Prof. M. Nussenzweig. CX₃CR1:GFP mice express EGFP in monocytes, dendritic cells, NK cells, and brain microglia under control of the endogenous cx3cr1 locus[39]. All mice were bred in the animal facility of the DRFZ. All animal experiments were approved by Landesamt für Gesundheit und Soziales, Berlin, Germany, in accordance with institutional, state, and federal guidelines.

**Surgical preparation for longitudinal intravital imaging.** We used a lateral approach to expose the femoral shaft of the right hind limb of mice anaesthetized with 1.5–2.0% isoflurane. Initially, an incision of ~1.5 cm was made into the shaved and disinfected skin between the knee and hip joint, parallel to the femur. The underlying muscles were dissected and retracted along the delimiting fascia. Then, a 0.65 mm pilot hole in the distal half of the diaphysis was drilled through the cortex using an electric precision drill (FBS 240/E Proxxon GmbH, Foehren, Germany) mounted on a stand to ensure minimal damage to the underlying bone marrow tissue. Ring forceps were applied to fix the bone in position. We placed the TiCP grade 4 implant parallel onto the femoral shaft and drilled the other two pilot holes (0.31 mm) for the bicortical screws manually through the complete shaft. Here, the holes in the implant act as drill templates. When fully inserted, the bone screws lock in the fixation plate, and thus prevent any movement or rotation. The wound was frequently washed during surgery with sterile NaCl solution and sewed with an absorbable surgical thread (Surgicryl Rapid PGA, USP 6-0, SMI, St. Vith, Belgium). Finally, we attached the reference plate. All animals were kept on a temperature-controlled heating device throughout the surgery (Supplementary Fig. 2). For longitudinal calvarial imaging we used a permanent window glued with bone cement onto the calvarial bone, leaving a circular window freely accessible[40]. The window glass was supported by a customized metal ring kindly provided by Dr. Rinnenthal, J.-L. (Charite—Universitätsmedizin, Berlin, Germany).

**Clinical scoring system.** Total clinical scoring post-surgery was based on behavior and appearance of the individual mice. The total clinical score represents the sum of eight factors (physical appearance, state of mucosae, motoric ability, wound healing, body weight loss, food and water consumption, response to provocation) rated on a scale of 0–3. No animal was scored to a total clinical score higher than 4.

**Histochemistry.** Femurs were fixed in 4% paraformaldehyde and transferred to 10–30% sucrose/PBS for cryoprotection. Fixed bone sample and implant were covered with Kawamoto's medium (SCEM, Section-Lab Co. Ltd., Hiroshima, Japan) in order to avoid air bubbles during separation. Bones were frozen and cryo-

sectioned using Kawamoto's film method[41]. Movat's pentachrome stainings were performed on 7 μm sections[42]. Brightfield images were generated by three-dimensional tile scanning on a Biorevo (BZ-9000, Keyence GmbH, Neu Isenburg, Germany) using the BZ-II Viewer with a 10×, NA 0.45 (air) objective lens. Images were stitched using BZ-II Analyzer and processed with Fiji software.

**Fluorescence microscopy**. Bones were fixed and sectioned as described above. Sections of 7 μm were blocked with 5% FCS/PBS for 30 min and stained with antibodies in 5% FCS/PBS/0.1% Tween for 1–2 h at room temperature: CD45 (1:100, ThermoFisher eBioscience, Frankfurt, Germany, 30-F11), Sca-1-APC (1:200, eBioscience, 17-5981-82), Laminin (1:200, Sigma-Aldrich, Taufkirchen, Germany, L9393), c-kit-PE (1:100, 130-102-542, Miltenyi, Bergisch Gladbach, Germany), Ki-67-bio (1:100, eBioscience, 14-5698-82), Endomucin (1:100, Santa Cruz, sc-65495), CD31-A488 (1:100, R&D Systems, FAB3628G). Sections were washed with PBS/0.1% Tween three times and incubated with secondary Abs (LifeSciences–Sigma-Aldrich, Taufkirchen, Germany) for 1 h at room temperature. Nuclei were stained with 1 μg ml$^{-1}$ DAPI (Sigma-Aldrich) in PBS and mounted with Fluorescent Mounting Medium (DAKO, Hamburg, Germany). Overview images were generated by three-dimensional tile scanning on a LSM710 (Carl Zeiss MicroImaging, Jena, Germany) using Zen 2011 software, with either a 10×, NA 0.3 (air) objective lens for complete femoral bone marrow sections, or a 20×, NA 0.5 (air) objective lens for partial scans (implant area, distal bone). Maximum intensity projection images and stitched overviews were created using Zen 2011 and processed with Fiji. For tracking of CMTPX-labeled transferred cells, splenocytes were isolated and incubated with CellTracker Red CMTPX, according to the manufacturer's protocol (ThermoFisher, Frankfurt, Germany). $1 \times 10^6$ splenocytes were transferred to recipient mice. Bones were harvested 4 h after transplantation and femurs were processed as described in the immunofluorescence method section. Sections were stained with DAPI and fluorescent images were generated by three-dimensional tile scanning on a Biorevo (BZ-9000, Keyence GmbH, Neu Isenburg, Germany) using BZ-II Viewer with a 10×, NA 0.45 (dry) objective lens. Images were stitched using BZ-II Analyzer and processed with Fiji software.

**EdU labeling and staining**. Proliferating cells were labeled using EdU at 1.6 mg kg$^{-1}$ mouse weight injected in PBS intraperitoneally 2 h before sacrifice[7]. Femoral bone sections were stained using the Click-iT EdU Alexa Fluor 647 kit (Life-Sciences–Sigma-Aldrich, Taufkirchen, Germany), according to the manufacturer's protocol.

**Flow cytometry**. Diaphyseal ends including growth plate and secondary ossification centers of femurs were cut off and discarded. An 18 gauge needle was inserted into the distal end of the long bone and the bone marrow was thoroughly flushed out with 5 ml of cold (4 °C) MACS buffer (0.5% BSA, 2 mM EDTA in PBS). LIMB implants were removed after flushing. Cells were re-suspended, filtered with a 50 μm cell strainer, spun down 6 min at $75 \times g$, and re-suspended in an erythrocyte lysis buffer. After washing, FcR were blocked using antibodies against CD16/32 (DRFZ in house clone 2.4G2, 5 μg ml$^{-1}$) and stained with antibodies on ice for 40 min. After washing the cells were acquired with a FACSymphony (BD Bioscience, Heidelberg, Germany) system and populations analyzed in FlowJo10. Cell counts were performed on $n = 6$ individuals. Live lymphocytes were divided into the following populations: progenitors: LSK cell (lin$^-$sca1$^+$ckit$^+$), long-term hematopoietic stem cell (LT HSC, lin-sca1$^+$ckit$^+$CD150$^+$CD48$^-$), short-term HSC (ST HSC, lin$^-$sca1$^+$ckit$^+$CD150$^+$CD48$^+$), multi-potent progenitor (MPP, lin$^-$sca1$^+$ckit$^+$CD150$^-$CD48$^+$), common lymphoid progenitor (CLP, LSK$^-$IL7R$^+$Flk2$^+$), common myeloid progenitor (CMP, lin$^-$sca1$^-$ckit$^+$CD34$^+$CD16/32$^-$), megakaryocyte-erythroid progenitor (MEP, lin$^-$sca1$^-$ckit$^+$ CD34$^-$CD16/32$^-$) and granulocyte-macrophage progenitor (GMP, lin$^-$sca1$^-$ckit$^+$CD34$^+$CD16/32$^+$), dendritic cell progenitor (DCP, lin$^-$sca1$^-$Flk2$^+$IL7R$^+$) B cells: propreB (B220$^+$CD19$^-$ckit$^+$Flk2$^+$), preBI (B220$^+$CD19$^+$ckit$^+$), preBII (B220$^+$CD19$^+$ckit$^-$IgM$^-$CD25$^+$), immature B cells (B220$^+$CD19$^+$ckit$^-$IgM$^+$IgD$^-$), mature B cells (B220$^+$CD19$^+$ckit$^-$IgM$^+$IgD$^+$), plasma cells (CD138high) T cells (CD3$^+$CD4$^+$), CD8 T cells (CD3$^+$CD8$^+$) innate immune cells: natural killer cells (NK cells, NK1.1$^+$), granulocytes (CD11b$^+$Gr1$^+$), monocytes/macrophages (CD11b$^+$Gr1$^-$), dendritic cells (DCs, CD11c$^+$). Antibodies used: CD19-FITC (1:800, BioLegend, Fell, Germany, 115506), B220-FITC (1:800, BioLegend 103206), Gr1-FITC (1:800, BioLegend 108406), CD3-FITC (1:800, BioLegend 100204), CD11b-FITC (1:800, BioLegend 101206), Ter119-FITC (1:800, BioLegend 116206), IL7R-Pe-Cy7 (1:100, BioLegend 135014), B220-BV510 (1:400, BioLegend 103247), CD34-A647 (1:200, eBioscience 51-0341-82), CD16/32-A450 (1:800, eBioscience 48-0161-82), sca1-APC-Cy7 (1:400, BioLegend 108126), Flk2-Pe (1:100, BioLegend 135306), CD19-BV650 (1:400, BioLegend 115541), ckit-A700 (1:200, eBioscience 56-1172-82), c-fms-APC (1:200, eBioscience 17-1152-82), ckit-APC (1:200, BioLegend 135108), CD150-Pe (1:800, BioLegend 115904), CD48-Pe-Cy7 (1:800, BioLegend 560731), sca1-A700 (1:200, eBioscience 56-5981-82), CD138-BV421 (1:800, BioLegend 142508), IgM-APC (1:400, BioLegend 406509), CD25-APC-Cy7 (1:400, BioLegend 102026), IgD-A488 (0.7 μg ml$^{-1}$, in house, clone 11.26c), CD4-Pe-Cy7 (1:800, eBioscience 25-0042-82), CD8-APC (1:800, BioLegend 100712), CD3-APC-Cy7 (1:200, eBioscience 47-0031-82), CD11c-FITC (1:800, BioLegend 117306), NK1.1-Pe (1:800, BioLegend

108707), Gr1-PB (1:800, BioLegend 108429), CD11b-A700 (1:800, eBioscience 56-0112-82).

**Mouse activity analysis**. Analysis of mouse activity and motility was performed using radio-frequency identification (RFID) technology to identify and track individual mice within a cage. Animals were subcutaneously tagged with bio-compatible glass-encapsulated RFID transponders (EURO I.D., Frechen, Germany) and were detected by a 2 × 4 sensor plate system (Phenosys, Berlin, Germany) underneath the cage. The frequency of crossed sensors serves as an approximate activity measure. Physical activity was normalized to the mean value of activity measured over 3 days before surgery. Before that, mice had been placed in the cage for at least 2 days. The relative change in activity for each mouse is displayed. For sham treatment, animals were put under anesthesia.

**Ex vivo μCT**. Femoral bones were harvested and fixed as described above. Fixed bones were measured in 20–30% sucrose solution using the vivaCT 40 (Scanco Medical AG, Bruettisellen, Switzerland). Scans were performed with 70 kVp and an isotropic voxel size of 10.5 μm. Reconstructed scans were converted into tiff-stacks for further data analysis. Bone morphological measurements from μCT 3D reconstructions were performed using the intensity. For bone thickness under the fixation plate and on the opposite site of the bone, areas were chosen as shown in Fig. 2e. For each position, the intensity profile was approximated with two Gaussian curves. The width of each Gaussian curve was considered to represent bone thickness at the respective position. For intact contralateral femurs, we performed the same measurements choosing a mirrored position of the bone to account for the fact that the LIMB femurs are always right femurs and the contralateral bones are left femurs.

**Data availability**. All relevant data are saved on the institutional servers. The data that support the findings of this study are available from the corresponding authors upon reasonable request.

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

## Acknowledgements

We thank Patrick Thiemann and Manuela Ohde for assistance with animal care. This work was supported by DFG FOR 2165 (NI1167/4-1 and NI1167/4-2 to R.A.N. and HA5354/6-1 and HA5354/6-2 to A.E.H.), TRR130, TPC01 (to R.A.N. and A.E.H.), TP17 (to A.E.H.) and TP16 (to H.D.C.) and HA5354/8-1 to A.E.H. D.R. and J.S. are members of the Berlin-Brandenburg School for Regenerative Therapies (BSRT). We would like to thank the Charité-Universitätsmedizin Electron Microscopy Facility (Prof. S. Bachmann) for help with experiments. The authors thank Randy Lindquist for proofreading of the manuscript.

## Author contributions

R.A.N., A.E.H., D.R. and J.S. designed the study, analyzed data and interpreted results. R.A.N., A.E.H., D.R., J.S., R.G., G.P., A.R. and S.Z. performed experiments. R.M., R.N., K.S.-B. and G.D. provided expertise for the implant design, surgery and Movat's pentachrome histochemical staining. F.M. provided expertise in the analysis of HSCs subsets frequencies and interpretation of these data with respect to the characterization of post-surgical homeostasis. Y.W., H.-D.C. and S.N. helped with mouse activity analysis. R.A.N. and A.E.H. wrote the manuscript.

## Additional information

**Competing interests:** R.M. and R.N. (RISystem AG, Davos, Switzerland) declare competing financial interests. The implant for longitudinal imaging will be commercialized by RISystem AG, Davos, Switzerland. The remaining authors declare no competing financial interests.

