## [Peer Review File · Nature Communications]

REVIEWERS' COMMENTS:

Reviewer #1 (Remarks to the Author):

The authors have greatly improved their manuscript and made it both more rigorous and clear. The work is overall impressive and worth of publication. The only point that I still find difficult to follow is the vascular remodelling shown in Figures 5-7. Could the authors please highlight with arrowheads and in 3D renders of the volumes observed the vascular remodelling observed. As it is, it could easily be an artefact of slight shifts in focus of the images. The immunofluorescence analyses presented do suggest there is vascular remodelling around the imaging window, therefore I believe it is just a matter of presenting the intravital findings in a more accessible way.

Reviewer #2 (Remarks to the Author):

The authors described a new method (LIMB) that enables longitudinal intravital imaging of the marrow of long bones. Using this method, the authors observed dynamic reorganization of the bone marrow vasculature, which seems to be an ongoing process even in the steady state after the marrow has recovered from the surgery. This observation has significant implications for the stability of different hematopoietic cell niches. The authors provided data excluding endothelial cell proliferation as the main driver of this reorganization but otherwise the mechanism for the vascular reorganization remains unclear. Overall the revised manuscript has been substantially improved; however my main concern remains that the imaging field of view is fixed and is very limited. The benefit of this technique is therefore unlikely to be broad-reaching in my opinion, when the benefit has to be balanced against the requirement for a rather invasive surgical procedure to implant and stabilize the GRIN lens. While it is true that even this small field of view will open a window into a region of the bone marrow that has not been accessible using previous imaging approaches, the manuscript as presented does not provide a compelling new finding that is unique to this region of the bone marrow. In fact all the results (cell motility, vascular reorganization, etc) observed using LIMB are also obtained using the more established intravital imaging of the calvarium, with no significant difference between these compartments. In addition, while the authors argue that the imaging depth can be adjusted from the endosteal to the deeper regions of the marrow, the need to drill through the bone makes it unlikely that the biology near the endosteum can really be studied because it has been replaced by a glass or sapphire interface.

Other concerns:

Lines 321-322: "both hematopoietic and stromal cells – dwell within the initially photo-activated tissue area over the entire period."

The hematopoietic cells are motile and move out of this area, as stated in line 343.

Line 392: "the imaged region remains stable over several repeated courses of photoactivation"
I cannot find data showing repeated photoactivation.

Lines 422-424: "stromal dynamics occurred on a slower time scale than the changes we observed in the vascular structure."

I cannot find any quantitative data in support of this statement.

Lines 648-649: "a big blood vessel (>100 μm diameter, possibly the main sinus)"
The indicated vessel appears to be smaller than the 100 μm scale bar.

Line 655: "Scale bar = 200 μm "

The field of view for the three-lens GRIN system is 150 μm so the scale bar cannot be 200 μm .

Lines 699-700: "general frequencies of CD45+ and Sca-1+ cells and expression of Lam is comparable between both the implanted and the contralateral femur. It will be helpful if these results are quantified."

Lines 705-706: "splenocytes ... engraft homogeneously 4 h after transplantation in both contralateral and LIMB implanted femurs"
The results need to be quantified.

Figure 6d: Shouldn't the differential image have both positive and negative values, indicating both the appearance and disappearance of blood vessels?

Suppl Figure 6: Leakage of Qdots seems to increase after photoactivation. Have the authors excluded the possibility that photoactivation may cause vascular damage?

Suppl Figure 7, lower panel: I believe the 0-7 min time lapse images are all taken 24 hours after photoactivation, but the box in the lower right panel gives the false impression that the 7 min image was obtained 48 hours after photoactivation.

Reviewer #3 (Remarks to the Author):

Intravital imaging the bone marrow (bm) is an important method to study key functions of the hematopoietic and immune system. While approaches for imaging a small patch of bm in the calvaria of mice have been developed a long time ago, imaging in long bones is far more difficult and therefore used only rarely, even though the marrow of long bones is considered much more essential for hematopoiesis and probably also the production of immune cells. The authors clearly make the important point, that hematopoietic processes take place over periods of days to weeks (although certainly not years in mice, as stated by the authors). Hence, previously available imaging approaches for the bm of long bones are only partially suitable to study such processes as they are typically one time-point terminal experiments.

The authors therefore developed an innovative new approach for the long-term intravital imaging of the bm in the femora of mice using a permanently attached metal support, a GRIN lens and time-lapse two photon imaging. They demonstrate the performance of their system by showing the migration of B cells in the bm over long periods of time. Furthermore, they show an unexpected plasticity of the bm vasculature.

This is a highly revised version of a paper that was previously submitted to Nature Methods. The authors have done a very comprehensive and convincing job to address all my concerns.

Point-by-point answer to reviewers' comments:

Reviewer #1 (Remarks to the Author):

The authors have greatly improved their manuscript and made it both more rigorous and clear. The work is overall impressive and worth of publication. The only point that I still find difficult to follow is the vascular remodeling shown in Figures 5-7. Could the authors please highlight with arrowheads and in 3D renders of the volumes observed the vascular remodeling observed. As it is, it could easily be an artifact of slight shifts in focus of the images. The immunofluorescence analyses presented do suggest there is vascular remodeling around the imaging window, therefore I believe it is just a matter of presenting the intravital findings in a more accessible way.

We thank the reviewer for the encouraging statements regarding our work. Following this suggestion, we emphasize in the revised version of the manuscript that all intravital fluorescent images of the vasculature (Fig.5-7) represent three-dimensional reconstructions. We additionally provide two movies showing the images in Fig. 6d during rotation around the y-axis to better emphasize the fact that vascular remodeling is a true biological phenomenon, not an artifact caused by inaccurate, repeated focusing. Furthermore, to make our point more clear, vessels that are changing were indicated with asterisks and stable vessels that are not changing and can be used as tissue landmarks with arrowheads.

Reviewer #2 (Remarks to the Author):

The authors described a new method (LIMB) that enables longitudinal intravital imaging of the marrow of long bones. Using this method, the authors observed dynamic reorganization of the bone marrow vasculature, which seems to be an ongoing process even in the steady state after the marrow has recovered from the surgery. This observation has significant implications for the stability of different hematopoietic cell niches. The authors provided data excluding endothelial cell proliferation as the main driver of this reorganization but otherwise the mechanism for the vascular reorganization remains unclear. Overall the revised manuscript has been substantially improved; however my main concern remains that the imaging field of view is fixed and is very limited. The benefit of this technique is therefore unlikely to be broad-reaching in my opinion, when the benefit has to be balanced against the requirement for a rather invasive surgical procedure to implant and stabilize the GRIN lens. While it is true that even this small field of view will open a window into a region of the bone marrow that has not been accessible using previous imaging approaches, the manuscript as presented does not provide a compelling new finding that is unique to this region of the bone marrow. In fact all the results (cell motility, vascular reorganization, etc) observed using LIMB are also obtained using the more established intravital imaging of the calvarium, with no significant difference between these compartments. In addition, while the authors argue that the imaging depth can be adjusted from the endosteal to the deeper regions of the marrow, the need to drill through the bone makes it unlikely that the biology near the endosteum can really be studied because it has been replaced by a glass or sapphire interface.

We thank the reviewer for acknowledging the importance of the observed vascular remodeling during steady-state homeostasis, especially in the light of how it may change our understanding of survival niche stability and function. We also agree that the mechanism driving this vascular remodeling (and the underlying stromal network remodeling) is unclear. However, the aim of this manuscript is to present this new longitudinal imaging technique, which allows us for the first time to image the deep marrow cavity of long bones, instead of being limited to the superficial cavity or to the isolated marrow islets of the calvarium. Finding out the underlying mechanisms of the newly observed vascular remodeling is the

subject of current research in our labs, which necessitates at least months to be finished, and, in our opinion, goes beyond the scope of this manuscript. As far as we are aware of, we are the first to identify this vascular remodeling as a physiologic phenomenon occurring in the steady state. Only using our new technique –taking advantage of the unique fixation of the lens within the bone—were we able to detect this reorganization and to quantify it properly. The well-established longitudinal calvarium imaging was used to confirm this observation, following the reviewers' suggestion to analyze whether this was a general phenomenon or restricted to the marrow of long bones. Taken together, we consider that properties of our system that were criticized by the reviewer (fixed and small imaging volume) in fact opened completely new insights into the biology of the bone and bone marrow, complementary to the perspective provided by longitudinal calvarium imaging. In order to address the reviewer's concerns, we added a paragraph addressing these aspects to the Discussion of our manuscript. We hope that this helps to make our aim and our perspectives better accessible for a broad readership.

Referring to the cellular motility of B lymphocytes, we completely agree with the reviewer that from this point of view, LIMB does not bring any new insight for the biology. Nevertheless, this comparative experiment was important for us to confirm that results (i.e. motility parameters of B cells and plasma cells) achieved using established imaging techniques of the bone marrow – terminal tibial imaging and longitudinal calvarium imaging – are similar to the results obtained by LIMB. Hence, we additionally confirmed the reliability of LIMB to acquire physiologic (and pathologic) information. This aspect is now better stated in the revised version of the manuscript.

Referring to the surgical burden for the animals when using LIMB, it is true that the process of implanting the microendoscope is more invasive than the surgical preparation for imaging in the calvarium. However, it does not impose a higher degree of burden to the animal than when tibial imaging is performed. In addition LIMB allows imaging at multiple time points in one and the same animal (until now, we have imaged up to 13 times, over the course of 115 days). We want to emphasize that these imaging sessions do not mean any further surgical burden to the animals, as the animals only receive a mild anesthesia for each imaging session. Thus, in our opinion, the balance between surgical burden and the benefit for imaging is clearly in favor of LIMB. Moreover, we are markedly reducing the number of animals while increasing the statistical accuracy, by avoiding inter-individual variance during the time-course of imaging. This aspect is emphasized in the revised manuscript.

Referring to the fact that the GRIN lens or the sapphire window cannot be put on a level with the endosteum, we agree also in this point with the reviewer. For this reason, we replaced in the manuscript "endosteal areas" with "pericortical areas". However, the various designs of LIMB allow us to reach various tissue areas, including endosteal areas. By using a very long endoscopic tubing, we can image the endosteum on the opposite side of the femur. By using very short endoscope tubings, bone and soft tissue grows from the cortical bone in front of the lens and forms endosteal areas de novo, which can be imaged (Figure for the reviewers is attached). A paragraph was added to the manuscript to answer this concern of the reviewer.

Finally, we agree with the reviewer that the main drawback (but, as discussed above, also the unique novel benefit) of LIMB is the fixed, rather small imaging volume. In order to clarify how

we are dealing with this as a drawback, we included in the revised version of the manuscript additional strategies we already developed to modify and enlarge the field of view. They include the use of a small prism (cathetus of 300 μm), glued at the end of the GRIN lens, which allow a side view of the tissue at various depths, and the fact that the length of the tubing is not limited to 500 and 700 μm , but can be adjusted to any length as needed for a certain application.

It is also true that we did not yet show many novel applications of LIMB involving the deep marrow of long bones – except for the vascular reorganization in the homeostasis – but taking into account that this is the first manuscript where this technique is presented, our aim is to focus on the technique and believe that further additional applications would go far beyond the aim of the present manuscript. Nevertheless, we completely agree with the reviewer that continuing our work to answer unprecedented questions using LIMB is absolutely necessary. Our labs are currently intensively pursuing this goal. We make this point clear in the revised version of the manuscript.

Other concerns:

Lines 321-322: "both hematopoietic and stromal cells – dwell within the initially photo-activated tissue area over the entire period." The hematopoietic cells are motile and move out of this area, as stated in line 343.

We thank the reviewer for making us aware about this misstatement in the manuscript. We changed the text of the manuscript indicating that only some photoactivated cells (presumably of stromal origin) persist over the entire period of 36 hours, whereas highly motile photoactivated cells, presumably hematopoietic cells, move out of the field of view earlier, as indicated also in our supplemental movies generated using paGFP mice (Fig. 5 and 7).

Line 392: "the imaged region remains stable over several repeated courses of photoactivation" I cannot find data showing repeated photoactivation.

We apologize for this mistake in phrasing. Repeated imaging sessions and not photo-activation sessions were meant in this context. The text of the manuscript was changed accordingly. We also performed repeated photo-activation sessions, but since we considered the added value of these experiments to be rather limited for the manuscript, we didn't refer to them in here.

Lines 422-424: "stromal dynamics occurred on a slower time scale than the changes we observed in the vascular structure." I cannot find any quantitative data in support of this statement.

As correctly indicated by the reviewer, a thorough quantification of our initial observations regarding the different dynamics of stromal versus vascular re-organization is necessary. Since this is subject of ongoing research in our labs and goes beyond the scope of the present manuscript. It is a middle-term perspective of ours to clarify this therefore we excluded this statement from the manuscript and will refer to this subject in future work.

Lines 648-649: "a big blood vessel (>100 μm diameter, possibly the main sinus)" The indicated vessel appears to be smaller than the 100 μm scale bar.

We agree with the reviewer that in xy the blood vessel is slightly smaller than 100 μm , however along the z axis it reaches between 96 and 204 μm . Even if accounting for the poor z-

resolution, inducing an error of 10 μm , the vessel is approximately 100 μm in width. Due to its dimensions and its location within the femur, it is very probable that the vessel is the main sinus. We corrected the statement in the revised version of the manuscript from ">100 μm " to "approx. 100 μm ".

Line 655: "Scale bar = 200 μm " The field of view for the three-lens GRIN system is 150 μm so the scale bar cannot be 200 μm .

We thank the reviewer for making us aware of this mistake, which occurred accidentally during editing. The correct scale bar is 100 μm , accordingly changed in the revised manuscript.

Lines 699-700: "general frequencies of CD45+ and Sca-1+ cells and expression of Lam is comparable between both the implanted and the contralateral femur.
It will be helpful if these results are quantified.

According to the previous suggestions of the reviewers, we decided to perform accurate and in-depth analysis of cell frequencies and numbers by flow cytometric analysis (Fig. 1 g) and to provide representative immunofluorescence histological data of at least $n = 3$ mice per group only in support of this flow cytometric quantification. Hence, the purpose of the imaging data was to provide an overview of the full length of the analyzed bones and of cellular localization in these bones, rather than to be the base of a thorough quantification. To make our intentions clear to the reader, we changed the description to: "Note the specific reaction to the implant-bone marrow interfaces indicated by accumulations of CD45+ cells, and Lam+ and Sca1+ arteries (yellow)."

Lines 705-706: "splenocytes ... engraft homogeneously 4 h after transplantation in both contralateral and LIMB implanted femurs"
The results need to be quantified.

We agree with the reviewer that our present data do not show the homogeneous distribution of the cells, but their presence both in the diaphyseal and metaphyseal regions of the femur, similar to distributions observed in contralateral bones. We changed the text of the revised manuscript accordingly, avoiding the indicated overstatement. The experiment the reviewer is referring to has been conducted following the advice of reviewer 1 after the initial submission, in order to provide proof that normal blood supply is maintained after implantation of the microendoscope. Reviewer 1 raised concerns about the blood flow in the femur because the reviewer was concerned that "the bicortical screw completely separates the left part of the bone marrow from the part under the knee". In the process of sectioning the bones for histology on a cryo-microtome we have never observed this type of separation. We used adoptive transfer of splenocytes to provide evidence of engraftment via the present vasculature and distribution throughout bone marrow of LIMB implanted femurs ($n = 3$ mice). Even though we did not quantify this result, the mere presence of splenocytes in the entire cavity suggests, that blood flow and engraftment are not massively hampered. We are convinced that the data is sufficient enough to dispel the concerns of the reviewer. In order to support the statement of homogeneous distribution of cells throughout the femoral bone marrow, accurate quantification of engraftment and distribution based on a higher number

of animals would be necessary in our opinion. Since such a statement goes beyond the scope of our present work and generating the necessary data would require at least 2-3 months, with marginal added value, we decided to exclude the unnecessary overstatement, rather than to perform this type of quantification.

Figure 6d: Shouldn't the differential image have both positive and negative values, indicating both the appearance and disappearance of blood vessels?

As indicated by the reviewers, the differential image between the 3D images of the vasculature acquired at a time distance of 24 hours contains both positive and negative values. In Fig. 6d, middle panel, only the positive values were displayed. For clarification, we replaced this image by a differential image containing the positive values in cyan and the negative values in yellow, the intensity of each color corresponding to the absolute value of the difference. We explain in the revised version of the manuscript how the normalization of the volume change was performed with respect to the total blood vessel volume. Additionally, we included a supplemental movie showing the rotating 3D reconstruction of the overlapped vasculature acquired at the time points 0 and 24 h, as well as a movie showing the rotating 3D reconstruction of the (negative in yellow and positive in cyan) differential image of the two time points. The movies reveal the changes in three dimensions and make clear that the vascular changes are not caused by focusing in different imaging planes.

Suppl Figure 6: Leakage of Qdots seems to increase after photoactivation. Have the authors excluded the possibility that photoactivation may cause vascular damage?

We thank the reviewer for pointing out the missing information. It takes about 40 min to accomplish paGFP activation as depicted here, using the specified 200 cycles. Leakage of the Qdots due to the fenestrated nature of the sinusoids in the bone marrow occurs naturally, independent of the photoactivation. Typically, in all mouse strains, independent of the imaging strategy of the bone marrow, i.e. in the femur by LIMB, in the tibia or calvarium, we observe after approx. 1 hour Qdots leaking out of the vasculature, due to the reason mentioned above. We added this central information to the revised manuscript. Additionally, the paGFP mouse strain has previously been used in in vivo two-photon-imaging experiments (Victoria et al, Cell, 2010). In this work, viability tests were performed, and no vascular damage was reported.

Suppl Figure 7, lower panel: I believe the 0-7 min time lapse images are all taken 24 hours after photoactivation, but the box in the lower right panel gives the false impression that the 7 min image was obtained 48 hours after photoactivation.

We thank the reviewer for pointing out this mistake and changed this misleading labelling accordingly in the revised manuscript.

Reviewer #3 (Remarks to the Author):

Intravital imaging the bone marrow (bm) is an important method to study key functions of the hematopoietic and immune system. While approaches for imaging a small patch of bm in the calvaria of mice have been developed a long time ago, imaging in long bones is far more difficult and therefore used only rarely, even though the marrow of long bones is considered much more essential for hematopoiesis and probably also the production of immune cells. The authors clearly make the important point, that hematopoietic processes take place over periods of days to weeks (although

certainly not years in mice, as stated by the authors). Hence, previously available imaging approaches for the bm of long bones are only partially suitable to study such processes as they are typically one time-point terminal experiments.

The authors therefore developed an innovative new approach for the long-term intravital imaging of the bm in the femora of mice using a permanently attached metal support, a GRIN lens and time-lapse two photon imaging. They demonstrate the performance of their system by showing the migration of B cells in the bm over long periods of time. Furthermore, they show an unexpected plasticity of the bm vasculature.

This is a highly revised version of a paper that was previously submitted to Nature Methods. The authors have done a very comprehensive and convincing job to address all my concerns.

We thank the reviewer for appreciating our work and pointing out the incorrect statement regarding processes taking place over years in mice. The statement was initially written to refer to humans and was not noticed when the text was edited. We excluded this statement from the revised manuscript.